# Multi-Frequency Electrocochleography and Electrode Scan to Identify Electrode Insertion Trauma during Cochlear Implantation

**DOI:** 10.3390/brainsci13020330

**Published:** 2023-02-15

**Authors:** Aniket A. Saoji, Madison K. Graham, Weston J. Adkins, Kanthaiah Koka, Matthew L. Carlson, Brian A. Neff, Colin L. W. Driscoll, Douglas C. Fitzpatrick

**Affiliations:** 1Department of Otolaryngology Head and Neck Surgery, Mayo Clinic, Rochester, MN 55905, USA; 2Department of Research and Technology, Advanced Bionics, Valencia, CA 91355, USA; 3Department of Otolaryngology-Head and Neck Surgery, University of North Carolina at Chapel Hill, Chapel Hill, NC 27599, USA

**Keywords:** cochlear implants, electrocochleography, cochlear microphonics, electrode scan

## Abstract

Intraoperative electrocochleography (ECOG) is performed using a single low-frequency acoustic stimulus (e.g., 500 Hz) to monitor cochlear microphonics (CM) during cochlear implant (CI) electrode insertion. A decrease in CM amplitude is commonly associated with cochlear trauma and is used to guide electrode placement. However, advancement of the recording electrode beyond the sites of CM generation can also lead to a decrease in CM amplitude and is sometimes interpreted as cochlear trauma, resulting in unnecessary electrode manipulation and increased risk of cochlear trauma during CI electrode placement. In the present study, multi-frequency ECOG was used to monitor CM during CI electrode placement. The intraoperative CM tracings were compared with electrode scan measurements, where CM was measured for each of the intracochlear electrodes. Comparison between the peak CM amplitude measured during electrode placement and electrode scan measurements was used to differentiate between different mechanisms for decrease in CM amplitude during CI electrode insertion. Analysis of the data shows that both multi-frequency electrocochleography and electrode scan could potentially be used to differentiate between different mechanisms for decreasing CM amplitude and providing appropriate feedback to the surgeon during CI electrode placement.

## 1. Introduction

In cochlear implants (CIs), an array of platinum electrodes is used to deliver electrical stimulation to the inner ear and restore hearing sensitivity and improve speech perception in patients with varying degrees of hearing loss. In patients with preserved postoperative acoustic residual hearing, combined acoustic and electrical stimulation can provide the best hearing outcome [1,2]. However, loss of residual hearing after cochlear implantation is an unintended consequence from the placement of the electrode array in the cochlea. Inner ear trauma due to electrode translocation from scala tympani to scala vestibuli is known to contribute towards this postoperative hearing loss [3].

Electrocochleography (ECOG) has been used with CI subjects to measure compound action potential (CAP), summating potential (SP), auditory nerve neurophonics (ANN), and cochlear microphonics (CM). CAP represents the onset and offset response of the auditory nerve, whereas SP is the direct current response from multiple generators. ANN reflects the phase-locking activity of the auditory nerve fibers, whereas CM represents the flow of electric current through the stereocilia of inner and outer hair cells and is used to monitor hair cell function during CI electrode placement [3,4,5,6,7,8,9,10,11,12,13,14,15,16]. A short-duration, alternating polarity, pure-tone stimulus such as 500 Hz, with a characteristic place of stimulation in the apical cochlear region, is typically used to generate CM. The most apical electrode from the implant array, with an extracochlear ground electrode, is used to measure CM during CI electrode placement. Advancement of the most apical electrode through the cochlear space leads to a gradual increase in CM amplitude due to the decrease in spatial separation between the site of CM generation, which is typically apical to the recording electrode location, and the recording electrode itself. A sudden decrease in CM amplitude during electrode placement is commonly associated with electrode insertion trauma or interference with inner ear basilar membrane mechanics. ECOG measurements are used to detect these events and provide timely feedback to the implanting surgeon to provide an opportunity to retract and reposition the electrode or limit further electrode advancement which may otherwise lead to an irreversible decrease in CM amplitude. However, a decrease in CM amplitude during CI electrode placement may be due to the advancement of the recording electrode beyond the site/s of CM generation in the cochlea or interaction between the sites as the array is moved [15,16]. Changes due to these factors can be misinterpreted as cochlear trauma, which can lead to unnecessary repositioning of the electrode, increasing the risk of causing cochlear trauma or an unwarranted partial insertion of the CI electrode. Therefore, it is of utmost importance to determine whether the decrease in CM amplitude is indicative of changes in basilar membrane mechanics or due to the advancement of the recording electrode through the cochlear space.

In the past, phase analysis of the recorded CM signal [7], comparison between intra- and extra-cochlear CM measurements [12], and/or multi-frequency electrocochleography measurements [14] have been used to differentiate between the two different mechanisms, leading to a decrease in intracochlear CM amplitude during CI electrode placement. Koka et al. used CM phase and amplitude changes to differentiate between changes in basilar membrane mechanics from advancement of the recording electrode beyond the site of CM generation on the basilar membrane [7]. The phase of the recorded CM signal is compared with that of the acoustic stimulus to calculate the phase change. Advancement of the recording electrode beyond the characteristic place of stimulation can be associated with a phase change of approximately 180 degrees [7,17,18]. In addition to phase analysis, Sijgers et al. compared intra- and extra-cochlear ECOG measurements to differentiate between the two mechanisms responsible for a decrease in CM amplitude during CI electrode placement. Their results show approximately 180 degrees phase shift and a decrease in intracochlear CM amplitude, without a similar decrease in extracochlear CM measurements during early stages of electrode placement. In that study, a correlated decrease in CM amplitude was measured on both intra-and extra-cochlear CM measurements, without significant changes in phase measurements during the latter half of electrode placement. Saoji et al. reported a single case study demonstrating the feasibility of using simultaneous presentation of multi-frequency tone bursts in measuring CMs from different locations along the basilar membrane during CI electrode placement [14].

In the present study, multi-frequency CM measurements performed during electrode placements were compared with amplitude and phase analysis of intraoperative electrode scan measurements performed after electrode placement. During multi-frequency electrode scan measurements, three or four frequency pure-tone bursts were presented, and the CM amplitude for each stimulus frequency was measured as a function of the intracochlear electrodes. If the decrease in CM amplitude during CI electrode placement is due to the advancement of the recording electrode, then peak amplitude measured during electrode placement should correlate with peak amplitude measured during electrode scan. If a decrease in CM amplitude during electrode placement is due to changes in basilar membrane dynamics or electrode insertion trauma, then the peak amplitude measured during electrode scan will be lower than the peak amplitude measured during electrode placement. Furthermore, phase changes consistent with a slowing down of the traveling wave as the characteristic frequency (CF) region is approached may provide an indication of electrode position within an atraumatic insertion.

## 2. Methods

In the present study, we report multi-frequency ECOG and electrode scan measurements from ten hearing-impaired patients (CI1 to CI10, 6 male and 4 female) with an average age of 73.1 years (SD = 10.4). In these patients, ECOG measurements were successfully measured during CI electrode placement (Table 1). Figure 1 (left panel) shows the preoperative audiogram for the ten CI patients. All patients were implanted with a HiRes Ultra 3D CI from Advanced Bionics, Valencia, California, USA. All patients underwent a standard mastoidectomy with facial recess and round-window electrode insertion at least to the first blue marker of the Advanced Bionics HiFocus SlimJ electrode array. A correlated decrease in CM amplitude at two more test frequencies was used to provide feedback to the surgeon. An attempt was made to preserve CM signal by retracting and repositioning the electrode when possible, but ultimately all patients received complete electrode insertion, which may be associated with a decrease in CM amplitude.

Multi-frequency electrocochleography measurements were performed using proprietary research software and hardware from Advanced Bionics, Valencia, CA, USA [19]. The research software was modified to perform multi-frequency ECOG measurements. Table 1 shows the frequencies of the acoustic-tone burst used to elicit CMs for each CI patient. The most apical electrode (electrode 1) was used as the recording electrode, along with the extracochlear case ground. The presentation level of the multi-frequency stimulus was set to 100 dB HL. Intraoperatively, multi-frequency CM tracings using three or four stimulus frequencies were performed for the ten CI patients. Following electrode placement, electrode scan or CM was measured as a function of the intracochlear electrodes. The electrode scan data were analyzed in terms of amplitude and phase changes as functions of the intracochlear electrodes. The study protocol was approved by Mayo Clinic Institutional Review Board (18–0088396).

Audiometric thresholds were measured using warble tones that were presented through a Madsen Astera dual-channel clinical audiometer with ER-3A insert earphones and a B-71 bone oscillator. Preoperative audiometric thresholds were measured within 1 month before CI surgery, and postoperative thresholds were measured within 1.5 to 2 months after implant surgery.

## 3. Results

Figure 1 shows pre- and post-operative air-conduction thresholds measured for the 10 CI patients. On average, the postoperative audiometric thresholds were 30 dB poorer than the preoperative thresholds, with a range of −5 to 95 dB. These results are discussed in relation to the intraoperative ECOG measurements in the Discussion section.

Figure 2 shows the multi-frequency CM tracings for the 10 CI patients. Electrode insertion time varied between 4 to 18 min across the 10 patients. All patients received complete electrode insertion up to the blue marker. A correlated decrease in CM amplitude at two or more frequencies was used to pause, retract, and/or reposition electrode placement for patients CI1, CI3, CI5, CI6, CI8, and CI10. For patients CI2 and CI7, a gradually increasing or stable CM amplitude was measured during electrode advancement, and therefore no adjustments were made during electrode placement. For patients CI4 and CI9, the CM amplitude was not used to optimize electrode placement due to rapid fluctuations in CM amplitude. In comparison to the peak CM amplitude measured during CI electrode placement, patients CI1, CI5, CI6, CI8, CI9, and CI10 showed significant decreases (>30%) in final CM amplitude measured for the different test frequencies. Patients CI3 and CI7 showed a small decrease (<30%) in CM amplitude during CI electrode placement. Patients CI2 and CI4 showed less than 30% decrease in CM for the lower test frequencies and greater than 30% decrease in CM for higher test frequencies (>1000 Hz), which may be attributed to the advancement of the recording electrode beyond the sites of CM generation for those frequencies. A greater than 30% decrease in CM amplitude measured for lower test frequencies such as 500 Hz has been shown to correlate with loss of residual hearing following CI surgery [9,13]. One advantage of using multi-frequency CM measurements is that cochlear trauma is likely to produce a correlated decrease in CM amplitude across the different test frequencies. For example, in several instances, patient CI6 showed a correlated decrease and recovery in CM amplitude measured for 253, 507, 761, and 1015 Hz. Moreover, patient CI8 showed a correlated decrease in CM for all the test frequencies in the early stage of CI electrode placement. On the other hand, patient CI5 showed a decrease in CM amplitude measured for stimulus frequency of 2011 Hz, followed by a decrease in stimulus frequency of 1015 Hz, and then 507 Hz, consistent with the tonotopic organization of the cochlea. While optimizing electrode placement, a decrease in CM amplitude at 1015 Hz was accompanied by an increase in CM amplitude at 507 and 253 Hz. This indicates the advancement of the recording electrode beyond the CM site/s of generation for 1015 Hz. Thus, in some patients, multi-frequency CM measurements can be used to differentiate a drop in CM amplitude associated with cochlear trauma from that produced by the advancement of the recording electrode beyond the site(s) of CM generation along the basilar membrane.

After completion of electrode placement, intraoperative electrode scan or multi-frequency CM measurements were performed as a function of the intracochlear recording electrodes. Figure 3 shows electrode scan measurements for the ten CI patients. For patients CI1, CI2, CI5, and CI8, electrode scan was measured on every alternate electrode from 1 to 15, due to limited time. Some cases showed a gradual increase in CM amplitude from the basal (number 16) to the apical electrodes (CI2, CI3, CI7). Others showed peaks prior to most apical electrode that then declined (CI1, CI6, CI10). Finally, some showed multiple peaks or dips during the sequence, suggestive of multiple sites for CM generation (CI5 for 507 and 1015 Hz, CI7 for 253 and 507 Hz, and CI10 for 253 and 525 Hz).

The analysis of phase was consistent with a slowing of the traveling wave at the characteristic frequency (CF) region for a given test frequency (Figure 4). The example shows the phase changes for the CM measured during electrode scan for CI7. The phase plot shows that in the base of the cochlea, the phase changed little for all frequencies. This region of shallow slope is consistent with a fast-traveling wave through these points or approaching the region with responding elements. Toward the apex, the slope increases, indicative of a slowing traveling wave as the CF region is approached. The association with different CF regions is apparent from the systematic locations where this slope change occurred (arrows). The CM amplitude plot for CI7 (Figure 2, CI7) shows the effect that summing regions of different phases can have. For 253 and 507 Hz, the dips in response at apical electrodes could be locations where the inputs of regions of different phases are interacting destructively. Thus, in addition to trauma and passing the region of maximal summation of response, a third cause of amplitude drops during insertion is interactions of source locations of different phases. Phase analysis revealed that patient CI5 showed phase reversal at 507 Hz near electrode 2 (Figure 2, CI5, filled symbol) and patient CI10 showed phase reversal at 725 Hz near electrode 2 (Figure 2, CI10, filled symbol). Phase reversal was not observed for the CM signal measured for the remaining 35 test frequencies across the 10 CI patients. These results are discussed in detail in the Discussion section.

To rule out electrode migration, after electrode insertion was complete, CM amplitude recorded at the end of electrode placement was compared with the CM amplitude measured during the electrode scan on electrode 1 (Figure 5, left panel). A correlation of 0.99 was obtained between the two different CM measurements for the 37 test frequencies across the 10 CI patients. This suggests minimal or no changes in electrode position after completion of electrode placement and the measurement of electrode scan. To differentiate between electrode insertion trauma from electrode advancement beyond the site(s) of CM generation as the cause for CM drops, the peak CM amplitude measured during electrode placement was compared with the peak CM amplitude measured during electrode scan. The results show a correlation of 0.39 (Figure 5, middle panel) between the peak CM amplitude measured during and after CI electrode placement. Note that patient CI8 showed large differences between the peak CM amplitude measured during electrode placement and the electrode scans (Figure 5, middle panel, black circles). This patient also shows a correlated decrease in CM signal for the four test frequencies near electrode 11, which may be suggestive of cochlear trauma or basilar membrane involvement (Figure 3, CI8, black circles). The peak CM amplitude from the tracings and the electrode scan showed an improved correlation of 0.85 when the data for patient CI8 were excluded from the analysis (Figure 5, right panel). These results suggests that the decrease in CM tracings observed during electrode placement can be at least partially and, in some cases, completely attributed to the advancement of the recording electrode beyond the multiple sites of CM generation along the basilar membrane.

To determine the relationship between intraoperative CM tracings and hearing preservation, the percent decrease in CM amplitude from the tracings were compared with the decrease in hearing across test frequencies ranging from 150 to 800 Hz across the 10 CI patients (Figure 6). Test frequencies beyond 800 Hz were not included in this analysis because the decrease in CM amplitude for high test frequencies such as 1000 and 2000 Hz may be due to the advancement of the recording electrode beyond the CM site/s of generation. The results show a lack of significant correlation (r = 0.07) between percent decrease in CM during CI electrode placement and decrease in residual hearing after cochlear implantation.

## 4. Discussion

Intra-operative CM monitoring is now being used to monitor hair cell function and guide intraoperative CI electrode placement. A decrease in CM amplitude is presumed to represent cochlear trauma and is used to provide feedback to the surgeon and alter the placement of CI electrode. Here, the intraoperative CM tracings combined with electrode scan measurements show that the decrease in CM amplitude may be due to the advancement of the most apical recording electrode beyond the site(s) of CM generation along the basilar membrane. The sites of maximal response are expected to be variable depending on the degree of hair cell preservation, which will affect the extent of the cochlea that responds. Low-frequency tones produce the best responses in CI subjects [20,21,22], which will, at the high intensities used here, excite most of the remaining hair cells, and the magnitude at a given location will be the vector average of the sum of all responses [23]. Thus, the region of maximal response at high intensities is expected to be well basal of the CF region according to a Greenwood map. Current electrodes in use for hearing preservation rarely advance as far as the 500 Hz CF region, so drops in response have been thought to represent possible trauma [24]. However, advancement of the recording electrode beyond the multiple site(s) of CM generation or a broadened region of CM generation due to loss of hair cells and poor frequency tuning can lead to a decrease in CM amplitude during CI electrode placement. This suggests that a decrease in CM amplitude alone should not be used to infer cochlear trauma and alter electrode placement. Either extracochlear CM measurement or electrode scan should be used in decision making during cochlear implantation.

The present study suggests that multi-frequency ECOG can be more reliable in detecting cochlear trauma as compared to the routinely used single-frequency ECOG measurements performed during CI electrode placement. Multi-frequency ECOG measurements are likely to show a simultaneous decrease in CM amplitude measured for the test frequencies during electrode insertion trauma, whereas a decrease in CM amplitude at one test frequency and a gradual increase in CM amplitude at other test frequencies are indicative of the advancement of the recording electrode beyond one of the multiple sites of CM generation or characteristic frequency along the basilar membrane [14].

Phase analysis of the electrode scans is consistent with the slowing of the traveling wave as the CF region is approached [25]. At basal sites, the increase in phase lag towards the apex was small, while at more apical sites the phase accumulation increased. This change in the slope of the phase change may be a more reliable indicator that the CF region is being approached than the amplitude changes. Koka et al. used this analysis to differentiate between cochlear trauma and the advancement of the recording electrode beyond the site of CM generation [7]. The reason for a lack of phase reversal in the present study may be two-fold. First, the temporal bone studies with the SlimJ electrode used in the present study have shown an average angular insertion of 413 degrees [24], which roughly corresponds to a place-pitch location of 750 Hz on the outer hair cell map. Therefore, it is unlikely that the most apical recording electrode will advance beyond the sites of CM generation for 250, 500, and possibly 750 Hz. Second, the lack of phase reversal for relatively higher frequencies such as 1000 and 2000 Hz may be due to higher audiometric thresholds, and consequently lower CM amplitudes, which may lead to unreliable phase analysis of the CM signal.

The lack of correlation between an intraoperative decrease in CM signal and postoperative residual hearing preservation is also because loss of residual hearing is a net result of intraoperative cochlear trauma (if any) and the foreign body response to the intracochlear electrode used to deliver electrical stimulation. To differentiate between intraoperative and delayed loss of residual hearing, Saoji et al. measured immediate postoperative bone-conduction thresholds that are representative of the inner hair cell function [13]. Their results showed delayed loss of residual hearing in patients with good intraoperative CM measurements and immediate postoperative residual hearing preservation. Furthermore, a decrease in CM amplitude is not indicative of cochlear trauma and is likely due to the advancement of the recording electrode beyond one or more sites of CM generation for a given stimulus frequency. One of the limitations of our study is the lack of pre- and post-operative imaging or CT-scan measurements showing intracochlear location of the CI electrodes in our study patients. Future studies should include imaging studies and correlate the estimated place-pitch location of intracochlear electrodes with intraoperative ECOG tracings and electrode scan measurements.

## 5. Conclusions

A low-frequency acoustic stimulus is commonly used to elicit CM and monitor hair cell function during CI electrode placement. A significant (>30%) decrease in CM amplitude is suggestive of cochlear trauma and is used for repositioning of the CI electrode. The present study indicates that advancement of the recording electrode beyond the different sites of CM generation can also lead to a decrease in CM amplitude and can be confused with electrode insertion trauma. Therefore, CM amplitude alone should not be used to make decisions influencing electrode placement and multi-frequency CM measurements, and/or a real-time implementation of the electrode scan can be used to differentiate electrode trauma from the advancement of the apical electrode beyond the CM source in the cochlea.

## Figures and Tables

**Figure 1 brainsci-13-00330-f001:**
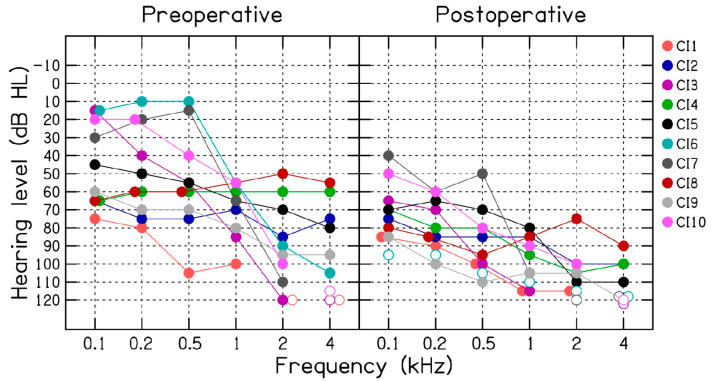
Pre and postoperative audiometric air-conduction thresholds (dB HL) measured for the ten (CI1 to CI10) cochlear implant patients.

**Figure 2 brainsci-13-00330-f002:**
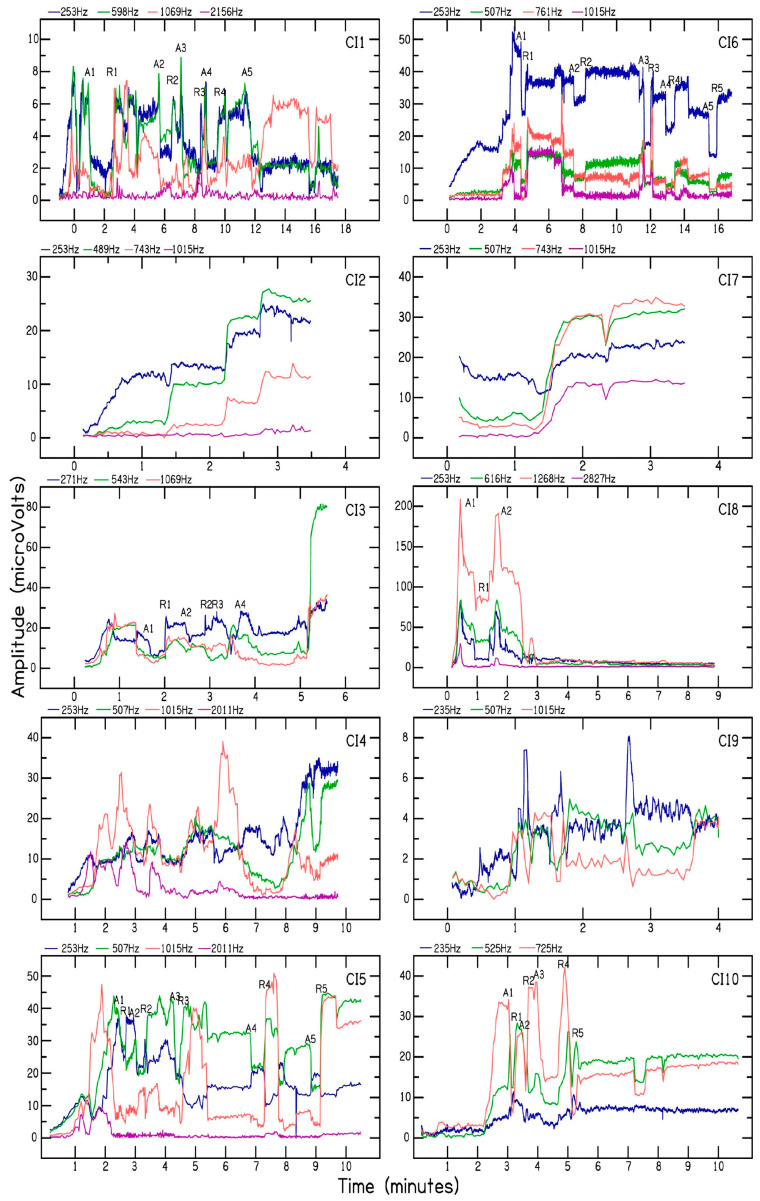
Intraoperative multi-frequency CM tracings measured for ten cochlear implant patients. The abscissa shows time (min) and the ordinate shows CM amplitude (µV). The frequencies used to elicit CMs are shown above each plot.

**Figure 3 brainsci-13-00330-f003:**
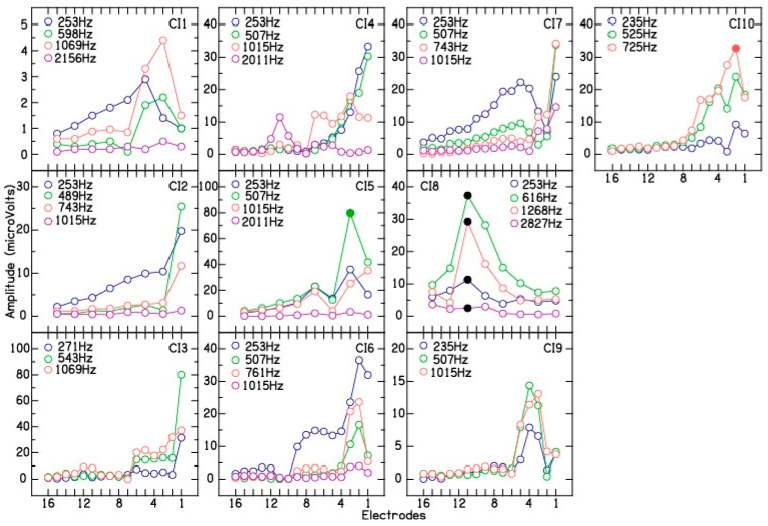
Intraoperative multi-frequency electrodes scan for the ten cochlear implant patients. The abscissa shows electrodes and the ordinate shows CM amplitude (µV). For patient CI8, black filled circles show correlated drops in CM amplitude for the four test frequencies. For patients CI5 (507 Hz) and CI10 (725 Hz), filled circles show phase reversal near electrode 2.

**Figure 4 brainsci-13-00330-f004:**
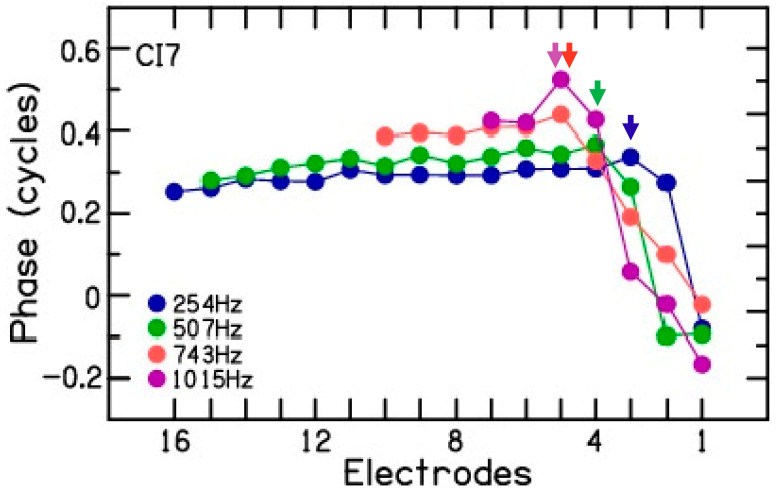
Example of phase in an electrode scan for patient CI7. The phase changes show two regions. In the basal cochlea, the phase changes are minimal, consistent with a fast—traveling wave through these regions or approaching the responding hair cells. In the apical region, the phase accumulation increases, consistent with a slowing traveling wave. There is a systematic relationship between the location stimulus frequency and electrode where the steeper slope begins.

**Figure 5 brainsci-13-00330-f005:**
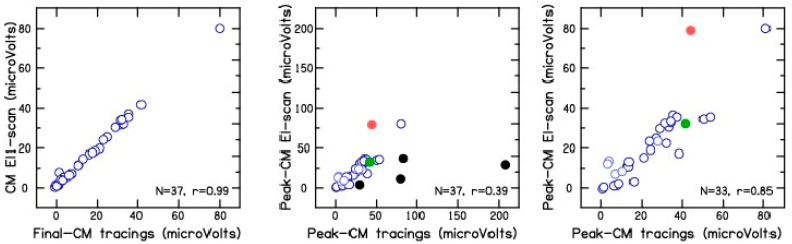
Left panel shows the correlation between the CM amplitude (µV) measured on the most apical electrode at the end of electrode placement and during electrode scan. Middle panel shows the peak CM amplitude (µV) measured during electrode placement and electrode scan. Black filled circles show correlated decreases in CM for the four test frequencies on electrode 11 for patient CI8. Green (CI5, 507 Hz) and red (CI10, 725 Hz) filled circles show phase reversal. Right panel shows the peak CM amplitude (µV) measured during electrode placement and electrode scan, but without the data for patient CI8. N refers to the number of test frequencies shown in each panel.

**Figure 6 brainsci-13-00330-f006:**
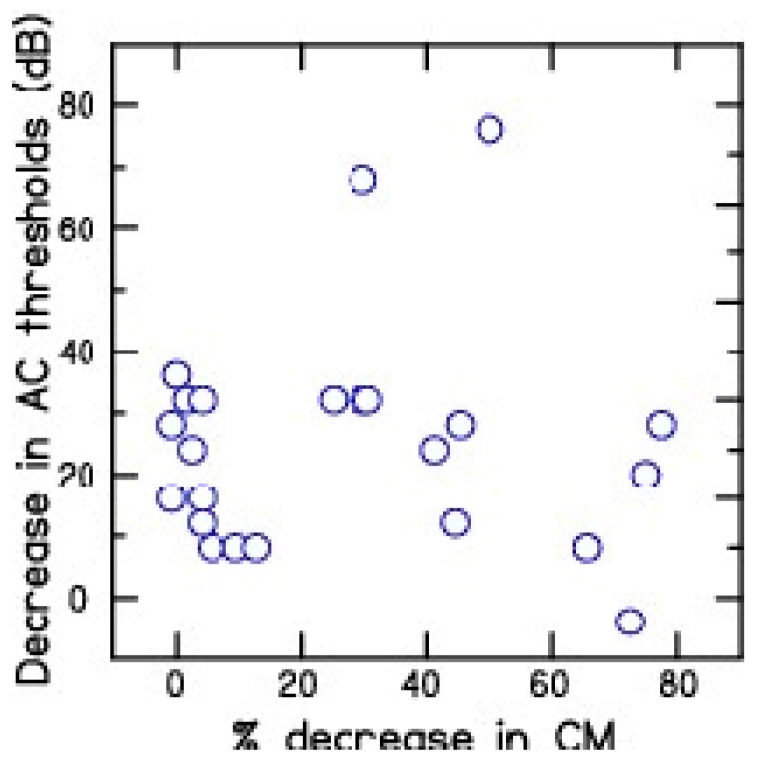
Comparison between percent decrease in CM (µV) amplitude and postoperative air-conduction thresholds (dB HL) across the frequency ranges of 250 to 800 Hz for the 10 cochlear implant patients.

**Table 1 brainsci-13-00330-t001:** Subject demographics, hearing thresholds and electrocochleography details.

Subjects	Age/Gender	Frequencies (Hz)	Decrease in CM (µV)	Preoperative Thresholds (dB HL)	Postoperative Thresholds (dB HL)
CI1	74/F	253, 598, 1069, 2156	82, 91, 71, 90%	75, 80, 105, 100, NR, NR	85, 90, 100, 115, 115, NR
CI2	76/M	253, 489, 743, 1015	13, 8, 17, 45%	65, 75, 75, 70, 85, 75	75, 85, 85, 85, 100, 100
CI3	87/M	271, 543, 1069	4, 1, 0%	15, 40, 55, 85, 120, NR	65, 70, 100, 115, NR, NR
CI4	54/M	253, 507, 1015, 2011	6, 0, 71, 95%	65, 60, 60, 60, 60, 60	70, 80, 80, 95, 105, 100
CI5	88/M	253, 507, 1015, 2011	56, 6, 29, 81%	45, 50, 55, 65, 70, 80	70, 65, 70, 80, 110, 110
CI6	73/F	253, 507, 761, 1015	38, 63, 83, 89%	15, 10, 10, 55, 90, 105	NR, NR, NR, NR, NR, NR
CI7	76/M	253, 507, 743, 1015	3, 0, 6, 6%	30, 20, 15, 65, 110, NR	40, 60, 50, 105, NR, NR
CI8	60/F	253, 616, 1268, 2827	94, 97, 98, 96%	65, 60, 60, 55, 50, 55	80, 85, 95, 85, 75, 90
CI9	72/F	235, 507, 1015	52, 38, 5%	60, 70, 70, 80, 95, 95	85, 100, 110, 105, 105, 120
CI10	71/M	253, 525, 725	39, 32, 57%	20, 20, 40, 55, 100, NR	50, 60, 80, 90, 100, NR

NR = no response. Subject demographics, intraoperative CM test frequencies, percent decrease in CM amplitude, pre-, and post-operative air-conduction thresholds measured for the ten (CI1 to CI10) cochlear implant patients.

## Data Availability

The data is not available due to privacy concerns.

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
