# Peer review of "Multi-Frequency Electrocochleography and Electrode Scan to Identify Electrode Insertion Trauma during Cochlear Implantation"

_brainsci, 2023, doi:10.3390/brainsci13020330_

Round 1

Reviewer 1 Report

Thanks for an interesting paper; I have a few minor and a few bigger comments.

Line 30:  There should be a brief discussion about ANN, CAP, and SP components of the ECoG, even though it’s not routinely utilized as part of intraop ECoG monitoring.

Line 38: I might suggest “between the site of CM generation, which is typically apical to the recording electrode location, and the recording electrode itself”

Line 44: I suggest “limit further electrode advancement which otherwise may lead to irreversible decrease”

Line 78-81: The sentence is pretty long and does not make it clear what is likely to be associated with the phase change. Is it’s the CM amplitude during electrode insertion, or CM amplitude during electrode scan?

Line 78-86 – Which of the two hypothesis refers to possible intraochclear trauma / loss of residual hearing?  That should also be made clear.

Lines 97 – 101: Can the authors show a figure of pre-op and post-op audiograms?  Did post-op audiograms show evidence of air-bone gaps?  EDIT: I see they present audiograms later in the paper.  Were there air bone gaps?  It may be worthwhile to show the audiograms earlier in the paper, but this is a minor comment / personal preference and I’ll leave it up to the authors.

Line 117:  Since there are 3-4 CM measurements going on during multi frequency ECoG, what were the criteria used to indicate a significant change that required manipulation of the electrode array?

Line 126: I assume the authors meant greater than 30% decrease?

Lines 216-226:  Data from other studies suggest that decrease in CM without any changes in phase may be more indicative of cochlear trauma rather than an advancement of the electrode array beyond the site of generation. The authors seem to suggest this too, if I understand correctly.  In addition, the authors saw that phase changes were not seen for 35 other test frequencies.  Did Figure 6 exclude data points of CM amplitudes with phase shifts? Would that improve the correlations?  What about instead combining all frequencies in one plot, perform 4 different correlations for the four different frequencies tested between 250 – 800 Hz. The point about the insertion depth the authors raise in line 265 is also a good point.

Line 279-280: “Their results … hearing preservation”  This seems to be an incomplete sentence. 

 Line 292: “CM cab ne” appears to be a typo.  I assume this is meant to be “CM can be"

Author Response

Line 30:  There should be a brief discussion about ANN, CAP, and SP components of the ECoG, even though it’s not routinely utilized as part of intraop ECoG monitoring.

Response: The revised version states that “Electrocochleography (ECOG) has been used to measure compound action potential (CAP), summating potential (SP), auditory nerve neurophonics (ANN), and cochlear microphonics (CM). CAP represents the onset and offset response of the auditory nerve whereas SP is the direct current response from multiple generators. ANN reflects the phase locking activity of the auditory nerve fibers whereas CM represents the flow of electric current through the stereocilia of the inner and outer hair cells and is used to monitor hair cell function during CI electrode placement”.

Line 38: I might suggest “between the site of CM generation, which is typically apical to the recording electrode location, and the recording electrode itself”

Response: Modified as suggested.

Line 44: I suggest “limit further electrode advancement which otherwise may lead to irreversible decrease”

Response: Modified as suggested.

Line 78-81: The sentence is pretty long and does not make it clear what is likely to be associated with the phase change. Is it’s the CM amplitude during electrode insertion, or CM amplitude during electrode scan?

Response: This section has been simplified as follows. “If the decrease in CM amplitude during CI electrode placement is due to the advancement of the recording electrode, then peak amplitude measured during electrode placement should correlate with peak amplitude measured during electrode scan. If a decrease in CM amplitude during electrode placement is due to changes in basilar membrane dynamics or electrode insertion trauma, then the peak amplitude measured on electrode scan will be lower than the peak amplitude measured during electrode placement. Furthermore, phase changes consistent with a slowing down of the traveling wave as the CF region is approached may provide an indication of electrode position within an atraumatic insertion”.

Line 78-86 – Which of the two hypothesis refers to possible intracochlear trauma / loss of residual hearing?  That should also be made clear.

Response: Please see above.

Lines 97 – 101: Can the authors show a figure of pre-op and post-op audiograms?  Did post-op audiograms show evidence of air-bone gaps?  EDIT: I see they present audiograms later in the paper.  Were there air bone gaps?  It may be worthwhile to show the audiograms earlier in the paper, but this is a minor comment / personal preference and I’ll leave it up to the authors.

We have included the pre- and post-operative audiometric thresholds in Table 1. Also, the pre and postoperative audiograms are included as Figure 1 in the Methods section. Measurable postoperative bone-conduction thresholds could not be obtained in some patients at the maximum limits of the audiometer, or the thresholds were influenced by vibrotactile sensation which made it difficult to estimate the air-bone gap for our implant patients. Therefore, we did not include bone conduction thresholds in our present study.

Line 117:  Since there are 3-4 CM measurements going on during multi frequency ECoG, what were the criteria used to indicate a significant change that required manipulation of the electrode array?

Response: This section has been modified as follows “A correlated decrease in CM amplitude at two or more frequencies was used to pause, retract, and/or reposition electrode placement for patients CI1, CI3, CI5, CI6, CI8, and CI10”.

Line 126: I assume the authors meant greater than 30% decrease?

Response: To improve clarity this section has been modified as follows “A correlated decrease in CM amplitude at two or more frequencies was used to pause, retract, and/or reposition electrode placement for patients CI1, CI3, CI5, CI6, CI8, and CI10. For patients CI2 and CI7 a gradual increase or stable CM amplitude was measured during electrode advancement and therefore no adjustments were made during electrode placement. For patients CI4, and CI9 the CM amplitude was not used to optimize electrode placement due to rapid fluctuations in CM amplitude. In comparison to the peak CM amplitude measured during CI electrode placement, patients CI1, CI5, CI6, CI8, CI9, and CI10 showed significant decrease (>30%) in final CM amplitude measured for the different test frequencies.  Patients CI3 and CI7 showed a small decrease (< 30%) in CM amplitude during CI electrode placement. Patients CI2 and CI4 showed less than 30% decrease in CM for the lower test frequencies and greater than 30% decrease in CM for higher test frequencies (>1000 Hz) which may be attributed to the advancement of the recording electrode beyond the sites of CM generation for those frequencies. A greater than 30% decrease in CM amplitude measured for lower test frequencies such as 500 Hz has been shown to correlate with loss of residual hearing following CI surgery [9, 13]”.

Lines 216-226:  Data from other studies suggest that decrease in CM without any changes in phase may be more indicative of cochlear trauma rather than an advancement of the electrode array beyond the site of generation. The authors seem to suggest this too, if I understand correctly.  In addition, the authors saw that phase changes were not seen for 35 other test frequencies.  Did Figure 6 exclude data points of CM amplitudes with phase shifts? Would that improve the correlations?  What about instead combining all frequencies in one plot, perform 4 different correlations for the four different frequencies tested between 250 – 800 Hz. The point about the insertion depth the authors raise in line 265 is also a good point.

Response: To improve clarity we have modified the discussion section as follows “Current electrodes in use for hearing preservation rarely advance as far as the 500 Hz CF region, so drops in response have been thought to represent possible trauma [21]. However, advancement of the recording electrode beyond the multiple site/s of CM generation or a broadened region of CM generation due to loss of hair cell and poor frequency tuning can lead to a decrease in CM amplitude during CI electrode placement. The result from our study suggests that advancement of the recording electrode beyond the multiple sites of CM generation, except the characteristic place, is unlikely to be associated with a phase reversal. This suggests that a decrease in CM amplitude alone should not be used to infer cochlear trauma and alter electrode placement. Either extracochlear CM measurement or electrode scan should be used in decision making during cochlear implantation”.

Phase analysis revealed that patient CI7 showed phase reversal at 507 Hz near electrode 2 (Figure 2, CI7, filled symbol) and patient CI10 showed phase reversal for 725 Hz near electrode 2 (Figure 2, CI10, filled symbol). Phase reversal was not observed for the CM signal measured for the remaining 35 test frequencies across the 10 CI patients. These results are discussed in detail in the discussion section.   

Figure 6 shows the comparison between intraoperative CM tracings and postoperative hearing preservation. To improve clarity, we have modified this section as follows “To determine the relationship between intraoperative CM tracings and hearing preservation the percent decrease in CM amplitude from the tracings and the decrease in hearing across test frequencies ranging from 150 to 800 Hz across the 10 cochlear implant patients (Figure 6). Test frequencies beyond 800 Hz were not included in this analysis because the decrease in CM amplitude for high test frequencies such as 1000 and 2000 Hz may be due to the advancement of the recording electrode beyond the CM site/s of generation. The results show a lack of significant correlation (r = 0.07) between percent decrease in CM during CI electrode placement and decrease in residual hearing after cochlear implantation”.

Thank you for your comment on the insertion depth and decrease in CM amplitude and phase changes.

Line 279-280: “Their results … hearing preservation”  This seems to be an incomplete sentence. 

Response:  Modified as follows “Their results showed delayed loss of residual hearing in patients with good intraoperative CM measurements and immediate postoperative residual hearing preservation”.

Line 292: “CM cab ne” appears to be a typo.  I assume this is meant to be “CM can be"

Response: Thank you for bringing this to our attention. This is corrected in the revised version.

Reviewer 2 Report

Advancements in cochlear implant surgical approaches and electrode designs have enabled preservation of residual acoustic hearing. Preservation of low-frequency hearing allows cochlear implant users to benefit from electroacoustic stimulation, which improves performance in complex listening situations, such as music appreciation and speech understanding in noise. Despite the relative high rates of success of hearing preservation, postoperative acoustic hearing outcomes remain unpredictable.

Hearing preservation is possible using soft surgical techniques and electrode arrays designed to minimize cochlear trauma; however, a subset of patients suffers from partial to total loss of acoustic hearing months to years following surgery despite evidence of residual apical hair cell function. Nowadays the advances in cochlear implantation have focused on minimizing cochlear trauma to improve hearing preservation outcomes, and in doing so expanding candidacy to patients with useful cochlear reserve. Recently, electrocochleography was introduced in the field of cochlear implantation since the cochlear implants received possibilities to measure intracochlear electrocochleography. 

Over the past few decades, there have been multiple applications of ECochG, with ongoing refinements in technique and updates in the understanding of recorded potentials. The ability to monitor cochlear trauma during CI electrode placement holds promise to improve hearing preservation outcomes and potentially modify surgical techniques and electrode design.

ECochG obtained through a cochlear implant is increasingly being tested as an intraoperative monitor during implantation with the goal of reducing surgical trauma. Reducing trauma should aid in preserving residual hearing and improve speech perception overall. Additionally, ECochG became a method of accurately assessing of electrode array placement in the cochlea. 

            ECochG measurements are used to detect these events and provide timely feedback to the implanting surgeon to provide an opportunity to retract and reposition the electrode or limit further electrode advancement which may lead to irreversible decrease in CM amplitude. However, decrease in CM amplitude during CI electrode placement may be due to the advancement of the recording electrode beyond the site/s of CM generation in the cochlea. This can be misinterpreted as cochlear trauma which can lead to unnecessary repositioning of the electrode increasing the risk of causing cochlear trauma or an unwarranted partial insertion of the CI electrode. Therefore, it is of utmost importance to determine if the decrease in CM amplitude is indicative of changes in basilar membrane mechanics or due to the advancement of the recording electrode through the cochlear space.

In the presented study, multi-frequency CM measurements performed during electrode placements were compared with amplitude and phase analysis of intraoperative electrode scan measurements performed after electrode placement.

The results of the presented study indicate that advancement of the recording electrode beyond the different sites of CM generation can also lead to a decrease in CM amplitude and can be confused with electrode insertion trauma. Therefore, CM amplitude alone should not be used to make decisions influencing electrode placement and multi-frequency CM measurements and a real time implementation of non-simultaneous multi-electrode electrode CM can be used to differentiate between electrode trauma from the advancement of the apical electrode beyond the CM source in the cochlea.To conclude, the presented article submitted for publication opens up new research capabilities in patients with cochlear implants and residual hearing, widening the existing theoretical and practical knowledge regarding ECochG in CI and further research progress in hearing preservation.

Author Response

Response: Thank you very much for your comments.

Reviewer 3 Report

I like to thank the Authors for the opportunity to review manuscript ”Multi-frequency electrocochleography and electrode scan to identify electrode insertion trauma during cochlear implantation”. As a topic, it would very well fit in electrocochleography special issue. The use of electrocochleography has also become area of interest in cochlear implant surgery and the research in order to understand the best way to exploit it during the insertion is under research. Thus, this paper would be valuable information in the field of cochlear implantation and electrocochleography and would provide foundation for larger research settings. There are some issues I would like to discuss.

Was this study retrospective or prospective and were there exclusion criteria for the patients? Were there only patients with good ECOG responses or did you exclude any where the ECOG did not provide good responses during insertion?

Demographics of the patients is missing, there is just age and gender. I would prefer the demographics to be shown in a table with pre- and postoperative hearing results. Or somehow fusing the table 1 with hearing results. As the measured CM frequencies are shown in table one, it would be nice to compare the frequencies and hearing results side by side.

Can you provide the information from preoperative imaging? The A- and B-measures would ad to the discussion regarding the electrode insertion depth, of course if you have postoperative imaging data (CT or CBCT) you could measure the insertion depth from those. For the postoperative measurements, the individual electrode sites could be estimated from postoperative scans (at least in most cases or make an “very accurate estimation of IDA” regarding the electrode length and contact distribution) and then correlated to the postoperative measurements. This could maybe provide better understanding regarding the areas where the CM is generated.   

All of the insertion were to the blue marker sign, so electrodes are fully inserted. The SLimJ is lateral wall electrode, and as mentioned, position approximated to 400 IDA. It is not very likely to control the advancement of the tip beyond the basal turn, even though the tip has been slightly bent. Were the main aim to insert the electrode fully or to achieve hearing preservation? Regarding the preoperative hearing threshold, patients 4, 6 and 10 would have been “candidates for EAS”. Now it seems that the low frequency hearing was lost (not useful for acoustic amplification anymore) in these patients. Did the surgeon made any adjustments to the insertion and what were the signs for the surgeon to do these repositions or adjustments of the electrode insertion vector? Were the “drops” in ECOG considered to be sign in these ten patients and did the more measured frequencies in ECOG influenced to the insertion in these cases? Did the authors considered partial insertion in order to preserve the residual hearing (Lenarz Otol Neurotol. 2019 Mar;40(3):e326-e335)? The insertion time is indicating that, the ECOG was taken in count as a tool regarding the insertion, so it was not measured as a research curiosity? Could this insertion protocol be opened bit more in methods?

After 270-300 degrees, the lateral wall electrodes tend to rise beneath the basilar membrane, so as full insertions of SlimJ, this will have some impact on ECOG measurements. It might have influenced also to the residual hearing results. Authors discuss about the foreign body reaction regarding the postoperative hearing results, this (as discussed) usually occurs later as the trauma will drop the hearing instantly. I was not able to find, when the postoperative hearing thresholds were gathered? This would probably give some idea whether the hearing deterioration occurred due the insertion related issues or foreign body reaction etc.

The approval for the study is mentioned twice.

There is both ECOG and ECochG (line 63) used in manuscript.

Author Response

Reviewer 3

I like to thank the Authors for the opportunity to review manuscript ”Multi-frequency electrocochleography and electrode scan to identify electrode insertion trauma during cochlear implantation”. As a topic, it would very well fit in electrocochleography special issue. The use of electrocochleography has also become area of interest in cochlear implant surgery and the research in order to understand the best way to exploit it during the insertion is under research. Thus, this paper would be valuable information in the field of cochlear implantation and electrocochleography and would provide foundation for larger research settings. There are some issues I would like to discuss.

Was this study retrospective or prospective and were there exclusion criteria for the patients? Were there only patients with good ECOG responses or did you exclude any where the ECOG did not provide good responses during insertion?

Response: In the revised version we state that “In the present study, we report multi-frequency ECOG and electrode scan measurements from ten hearing impaired patients (CI1 to CI10, 6-male and 4-female) with an average age of 73.1 years (SD=10.4). In these patients ECOG measurements were successfully measured during CI electrode placement”.

Demographics of the patients is missing, there is just age and gender. I would prefer the demographics to be shown in a table with pre- and postoperative hearing results. Or somehow fusing the table 1 with hearing results. As the measured CM frequencies are shown in table one, it would be nice to compare the frequencies and hearing results side by side.

Response: We have included the pre and post-operative audiometric thresholds in Table 1. Also, the pre and postoperative audiograms are included as Figure 1 in the Methods section.

Can you provide the information from preoperative imaging? The A- and B-measures would ad to the discussion regarding the electrode insertion depth, of course if you have postoperative imaging data (CT or CBCT) you could measure the insertion depth from those. For the postoperative measurements, the individual electrode sites could be estimated from postoperative scans (at least in most cases or make an “very accurate estimation of IDA” regarding the electrode length and contact distribution) and then correlated to the postoperative measurements. This could maybe provide better understanding regarding the areas where the CM is generated.

Response: Unfortunately, we did not obtain pre- or post-operative imaging studies or CT-scan for our study patients. We have included this limitation of our study in the discussion section. We are currently enrolled in a multi-center ECOG study which involves pre- and post-operative imaging, ECOG measurements in relation to hearing preservation and cochlear implant outcomes. This study will compare intraoperative ECOG measurements with insertion depth angle.

All of the insertion were to the blue marker sign, so electrodes are fully inserted. The SLimJ is lateral wall electrode, and as mentioned, position approximated to 400 IDA. It is not very likely to control the advancement of the tip beyond the basal turn, even though the tip has been slightly bent. Were the main aim to insert the electrode fully or to achieve hearing preservation? Regarding the preoperative hearing threshold, patients 4, 6 and 10 would have been “candidates for EAS”. Now it seems that the low frequency hearing was lost (not useful for acoustic amplification anymore) in these patients. Did the surgeon made any adjustments to the insertion and what were the signs for the surgeon to do these repositions or adjustments of the electrode insertion vector? Were the “drops” in ECOG considered to be sign in these ten patients and did the more measured frequencies in ECOG influenced to the insertion in these cases? Did the authors considered partial insertion in order to preserve the residual hearing (Lenarz Otol Neurotol. 2019 Mar;40(3):e326-e335)? The insertion time is indicating that, the ECOG was taken in count as a tool regarding the insertion, so it was not measured as a research curiosity? Could this insertion protocol be opened bit more in methods?

Response: Thank you for your suggestions. We have made modifications at in the Methods and Results section to elaborate on your suggestions.

In the Methods section, we state that “In the present study, we report multi-frequency ECOG and electrode scan measurements from ten hearing impaired patients (CI1 to CI10, 6-male and 4-female) with an average age of 73.1 years (SD=10.4). In these patients ECOG measurements were successfully measured during CI electrode placement (Table 1). Figure 1 (left panel) shows the preoperative audiogram for the ten CI patients. All patients were implanted with an Advanced Bionics HiRes Ultra 3D CI. All patients underwent a standard mastoidectomy with facial recess and round window electrode insertion at least to the first blue marker for the Advanced Bionics HiFocus SlimJ electrode array. A correlated decrease in CM amplitude at two more test frequencies was used to provide feedback to the surgeon. An attempt was made to preserve CM signal be retracting and repositioning the electrode when possible but ultimately all patients received complete electrode insertion which may be associated with a decrease in CM amplitude”.

In the results section we state that “A correlated decrease in CM amplitude at two or more frequencies was used to pause, retract, and/or reposition electrode placement for patients CI1, CI3, CI5, CI6, CI8, and CI10. For patients CI2 and CI7 a gradual increase or stable CM amplitude was measured during electrode advancement and therefore no adjustments were made during electrode placement. For patients CI4, and CI9 the CM amplitude was not used to optimize electrode placement due to rapid fluctuations in CM amplitude”.

After 270-300 degrees, the lateral wall electrodes tend to rise beneath the basilar membrane, so as full insertions of SlimJ, this will have some impact on ECOG measurements. It might have influenced also to the residual hearing results. Authors discuss about the foreign body reaction regarding the postoperative hearing results, this (as discussed) usually occurs later as the trauma will drop the hearing instantly. I was not able to find, when the postoperative hearing thresholds were gathered? This would probably give some idea whether the hearing deterioration occurred due the insertion related issues or foreign body reaction etc.

Response: In the revised Methods section we state that “Preoperative audiometric thresholds were measured within 1 month before CI surgery, and postoperative thresholds were measured within 1.5 to 2 months after implant surgery”.

The approval for the study is mentioned twice.

Response: Thank you for bringing this to our attention! The first mention of study approval has been deleted from the revised version of the manuscript.

There is both ECOG and ECochG (line 63) used in manuscript.

Response: Thank you! ECochG has been modified to ECOG.